# Orientation-invariant autoencoders learn robust representations for shape profiling of cells and organelles

James Burgess [1] ✉, Jeffrey J. Nirschl [2], Maria-Clara Zanellati [3], Alejandro Lozano[4], Sarah Cohen [3] & Serena Yeung-Levy [5,6,7] ✉

Cell and organelle shape are driven by diverse genetic and environmental factors and thus accurate quantification of cellular morphology is essential to experimental cell biology. Autoencoders are a popular tool for unsupervised biological image analysis because they learn a low-dimensional representation that maps images to feature vectors to generate a semantically meaningful *embedding space* of morphological variation. The learned feature vectors can also be used for clustering, dimensionality reduction, outlier detection, and supervised learning problems. Shape properties do not change with orientation, and thus we argue that representation learning methods should encode this orientation invariance. We show that conventional autoencoders are sensitive to orientation, which can lead to suboptimal performance on downstream tasks. To address this, we develop O2-variational autoencoder (O2-VAE), an unsupervised method that learns robust, orientation-invariant representations. We use O2-VAE to discover morphology subgroups in segmented cells and mitochondria, detect outlier cells, and rapidly characterise cellular shape and texture in large datasets, including in a newly generated synthetic benchmark.

Microscopy is the foundation of modern cell biology. High-throughput image-based assays have become essential for genetic screens and discovery-based biological research[1], drug profiling[2], and creating cellular and subcellular atlases[3,4]. Accurately quantifying cell and organelle morphology (i.e., size, shape, or structure) is essential to characterising changes in structure and function as a result of genetic or environmental perturbation. Autoencoders are a popular unsupervised representation learning method that learn a mapping from images to a low-dimensional vector representations which form an *embedding space*. These representations are useful for clustering, dimensionality reduction, and outlier detection. The geometric and shape properties of cells and organelles do not change with orientation, and thus ideal feature representations should be orientation invariant. However, existing autoencoders do not guarantee orientation-invariant features. Thus, similar but rotated shapes may not be close in the embedding space, which introduces orientation as a confounding variable for downstream clustering or analysis tasks.

Classical approaches to learning shape and morphology representations without supervision include principal component analysis (PCA)[5–7], Fourier and spherical basis functions[5,8], and deformation techniques[9]; they are only able to model binary segmentation masks. More recently neural-network (NN) based autoencoders (AE)[10–12], also unsupervised, have been used to model arbitrary biological images including segmented shapes, greyscale images, and multi-channel

[1]Institute for Computational & Mathematical Engineering, Stanford University, Stanford, CA, USA. [2]Department of Pathology, School of Medicine, Stanford University, Stanford, CA, USA. [3]Department of Cell Biology and Physiology, University of North Carolina at Chapel Hill, Chapel Hill, NC, USA. [4]Department of Biomedical Data Science, Stanford University, Stanford, CA, USA. [5]Departments of Biomedical Data Science, Computer Science, and Electrical Engineering, Stanford University, Stanford, CA, USA. [6]Chan Zuckerberg Biohub - San Francisco, San Francisco, CA, USA. [7]Clinical Excellence Research Center, School of Medicine, Stanford University, Stanford, CA, USA. ✉e-mail: jmhb@stanford.edu; syyeung@stanford.edu

images[8,13,14]. Other works use autoencoders with additional self-supervised loss functions[15,16]. These approaches tend to be sensitive to orientation: rotating or flipping the image will change its position in embedding space[5,8,13], which may reduce the accuracy of analysis tasks like clustering. This can be mitigated by image 'prealignment' to a canonical pose using rigid body transformations[5,8]. However, we show that the image prealignment strategy fails to consistently produce similar feature representations for similar shapes, which poses problems for downstream analyses like clustering.

Given the limits of classical autoencoders, we develop a framework for modifying autoencoder models[10–12] to leverage recent advances in geometric deep learning[17–19]. We constrain the architecture of the neural-network encoder to guarantee that output representations are invariant to input image orientation. This obviates the need for prealignment and ensures related shapes have similar feature representations, which may improve downstream tasks that use these features. However, the orientation-invariant encoder causes the reconstructed image to be misaligned with the input, which we correct by searching for the optimal alignment in Fourier space[20,21]. In this paper, we develop and experiment with O2-VAE, which implements our orientation-invariance framework on the variational autoencoder (VAE)[11,12]. Our approach is compatible with other unsupervised and self-supervised autoencoder methods that extend the AE and VAE[8,13–16,22].

We use O2-VAE to learn shape representations on biological data for clustering, visualisation, outlier detection, and feature learning. First, we generate and publicly release a synthetic dataset of greyscale cells with systematically varying eccentricity, contour irregularity, and cytoplasm texture classes. Since shape datasets with precise, well-defined ground-truth labels are uncommon, this is a useful baseline for other researchers. Using this labelled dataset, we show how the learned embedding space organises cell images based on morphology: first into coarse shape followed by fine shape details and then texture. We continue to explore the learned embeddings on cell segmentations of Phalloidin-labelled cells[7] and discover that thin, F-actin-positive filopodia-like structures are reduced in lamin-A-deficient mouse embryonic fibroblasts (MEFs), suggesting roles for lamin outside of nuclear shape. In human induced pluripotent stem cells (hiPSCs)[23] we detect mitosis cells without supervision, identify outliers, estimate how cell shapes may deform, and rapidly profile shape variation in the large dataset. Next we generate a new dataset of hIPSCs with six tagged organelles[24] and train O2-VAE on mitochondria with more complex shapes. Using these learned shape representations, we identify mitochondrial morphology groups and quantify how shape correlates with inter-organelle contact rates. Moving beyond segmentation-only data, we take greyscale nuclei images undergoing mitosis[25], and show preliminary results that O2-VAE can be trained to model shape and texture. Finally, we show that compared to existing autoencoder models, O2-VAE representations better separate distinct shape classes for datasets with available labels. We release O2-VAE code and example applications at https://github.com/jmhb0/o2vae/.

## Results

### O2-VAE enforces orientation-invariant representations

We develop O2-VAE, an unsupervised representation learning method for biological images based on VAEs[10–12] that learns the same feature representation for all orientations of a given image (Fig. 1a). Conventional VAEs, similar to autoencoders (AEs), map images into a compressed vector representation using neural-network encoders, which is then decoded to reconstruct the original image. These encoders are sensitive to input image orientation. Instead, we use O(2)-*equivariant* convolutional layers. Equivariance in this model means that if the input to a convolutional layer is rotated or reflected, then the output to that layer will be equivalently rotated or reflected (standard convolutions do not have this property). By stacking O2-equivariant layers, followed by spatial pooling, the encoder is O(2)-invariant[18,19,26]: as in Fig. 1a, if the input image is rotated or reflected, then the output of the 'encoder' does not change (Fig. 1a). We then use a regular deconvolutional decoder[27] to decode the representation to an image. However, since the encoder maps all image orientations to the same representation

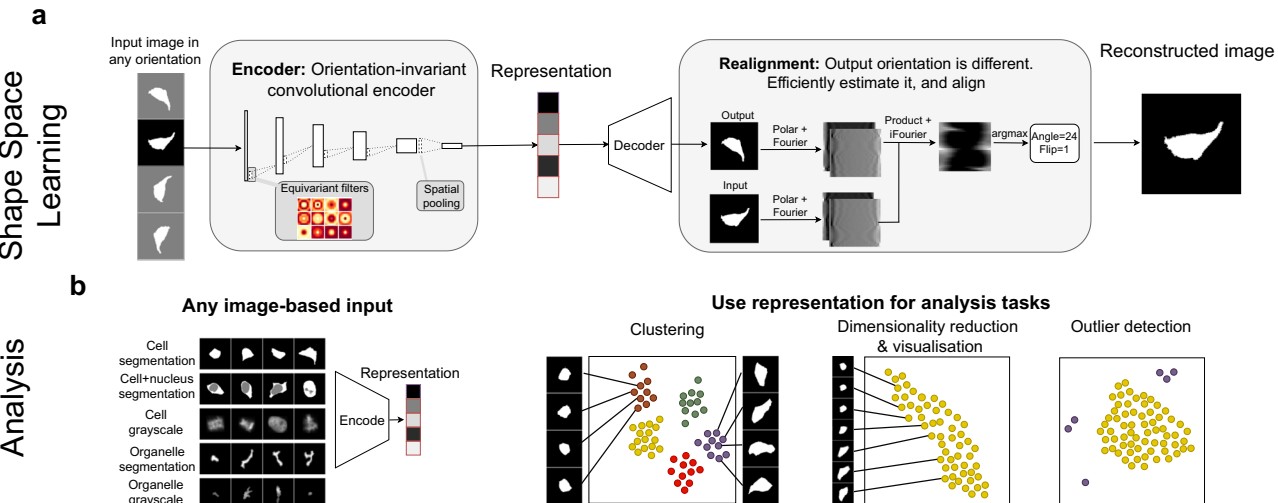

**Fig. 1 | O2-VAE learns orientation-invariant representations of cells and organelles. a** The O2-VAE model. An input image to the orientation-invariant encoder produces the same output vector for any input orientation. In the box labelled 'encoder' each layer of the convolutional encoder is constrained to be orientation-equivariant with a final spatial pooling layer to produce an orientation-invariant learned *representation* vector of cell phenotype. During training, the representation is decoded using a separate neural-network decoder, which is trained to reconstruct the input. By design, the learned representation is orientation invariant and thus the reconstruction orientation may differ from the input. In the box labelled 're-alignment', we efficiently estimate and correct the misalignment using Fourier transform-based methods. We use a loss function (not shown, for simplicity of the figure) that promotes accurate reconstruction while constraining the distribution of representations. **b** O2-VAE can be trained on any image including: binary, greyscale, and multi-channel images. The learned representation vector or phenotypic profile can be used for downstream analysis; three representative tasks are shown with each dot corresponding to an image in embedding space. For discretely-varying shapes, objects form separated clusters; for continuously-varying shapes, data can be visualised with dimensionality reduction; outlier data will be far from most other data.

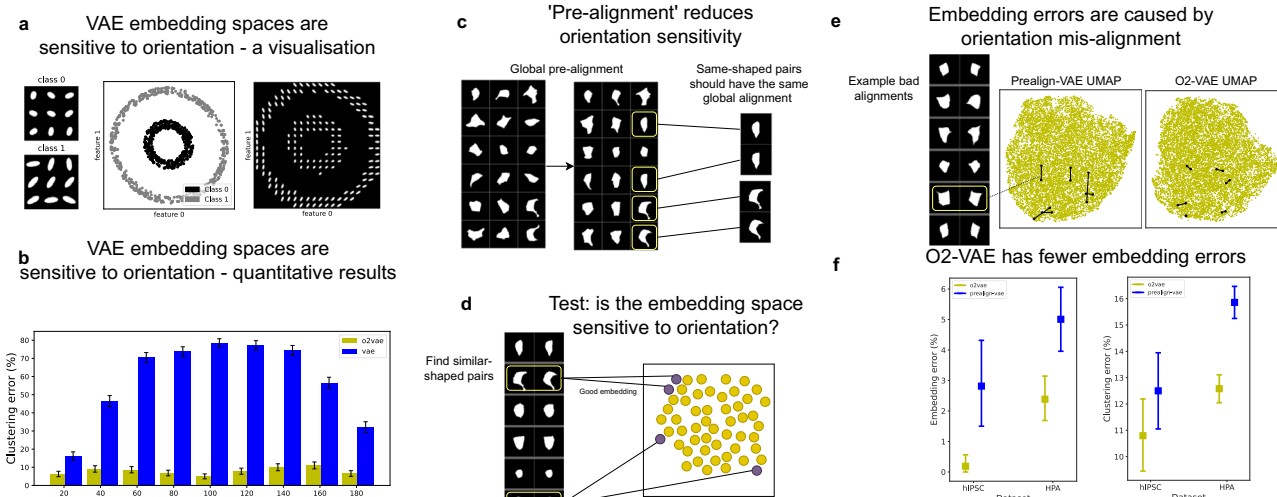

**Fig. 2 | Existing shape space methods are sensitive to orientation, which is not resolved by image prealignment. a** We train a VAE on a synthetic two-class dataset of ellipses (left panel). The classes separate in the embedding space forming nested circles (coloured by class, middle panel). Sampled images in the embedding space (right panel) show that orientation is encoded along the circumference of the nested circles, which means that objects with similar shape but different orientation are highly separated (right panel). **b** Clustering error rates from a synthetic cell dataset on O2-VAE vs VAE: an error means the cell and its rotated version are assigned to a different cluster. Absolute angles far from 0° and 180° have higher error, suggesting that orientation sensitivity is the issue (histogram height is mean value, and error bars are 95% confidence intervals, CI from nonparametric bootstrap). The remaining panels are about VAE with prealignment. **c** The result of a preprocessing algorithm to `prealign' 2d segmented hiPSCs. Given a dataset (left), rotate and flip images (middle) so that objects with similar shape are aligned with each other (right). This helps ensure they have similar learned representations.

**d** We design a test to check whether pairs of images with similar shapes have similar embeddings. We identify similar-shape pairs (Methods), then we identify `embedding errors', which is when a similar-shaped pair is separated in the embedding space. **e** Example image pairs that are `embedding errors' for prealignment-based methods but not for O2-VAE. They have bad pairwise alignment after prealignment. We show UMAPs for prealign-VAE and O2-VAE and draw lines between the embeddings of the example pairs that are visualised on the left. **f** Quantitative comparisons of errors with O2-VAE and prealign-VAE for two datasets: hiPSCs[23] and HPA[3]; (centre point is the mean value, and error bars are 95% confidence intervals, CI from nonparametric bootstrap). (Left) if similar-shaped pairs are far in embedding space, they are an `embedding error'. (Right) if similar-shaped pairs are not grouped by clustering, it is a `clustering consistency error'. Prealign-VAE has higher errors than O2-VAE. Source data for (**b**) and (**f**) are provided as a Source Data file.

vector, the output reconstruction may be misaligned with the input. We re-orient the input to align with the reconstruction, which can be done efficiently by transforming both images to polar-Fourier space, and interpreting their cross-correlogram[20] (Methods). The loss function has two terms: reconstruction error between re-oriented input and output, and a constraint on representations that encourage a smooth embedding space (this second loss term is the difference between the VAE and the autoencoder). The framework can be extended to enforce translation-invariant representations[28], enforce scale-invariant representations[29], and to model invariance in 3d shapes[30,31]. It can also be integrated with other VAE models.

### Image prealignment fails to enforce orientation-invariant embedding spaces

Existing representation learning methods are sensitive to image orientation: rotating or flipping the image changes its representation[5,7–9,13,32–34]. As an example, we train a VAE[5,8,11,12] on a synthetic dataset with two distinct classes (Fig. 2a). The representations form nested circles in embedding space (Fig. 2a middle), and the location in embedding space corresponds to both shape *and* orientation (Fig. 2a right): they are *sensitive to image orientation*. Naive *k*-means clustering fails in this simple example because many class 1 shapes are far from each other, but close to class 0 shapes. To investigate this clustering failure on more realistic data, we generated a synthetic dataset using the SimuCell software[35]: we created cell subpopulations by sampling from parameterised distributions with a known population mean eccentricity and contour randomness (Fig. 3a). We release this dataset as the Profiling Cell Shape and Texture (PCST) benchmark. For each image in PCST, we make a randomly rotated copy and test whether they are clustered together in a VAE embedding space. For *k*-means clustering with $k = 10$, the VAE has 58%

error rate, compared to O2-VAE with 7.4% error rate. Moreover, the error is worse when the rotation is far from 0° and 180° (Fig. 2b), supporting the claim that the errors are due to orientation sensitivity (more details in Supplementary Note 2h).

One solution to orientation sensitivity is to *prealign* input images to a canonical orientation, as demonstrated for cell data in Fig. 2c. Prealignment is the standard approach for limiting orientation sensitivity of representations[5,8] (Methods). We test whether pre-aligning images before training a VAE, called 'prealign-VAE', reduces orientation sensitivity (Fig. 2d). We use an 'embedding error' score: if the error score is high, then the embedding method frequently fails to correctly identify when two objects are similar (probably due to different orientation). To compute the embedding error score, we first identify pairs of images that are similar by pairwise aligning their orientations[20] and then computing a normalised similarity score in pixel-space. We choose a strict threshold for the pixel-space similarity score and we visually verify that they are similar (Supplementary Note 1c), so these pairs should be very close in embedding spaces for shape profiling. To test that these pairs really are close in the cell profiling embedding space, we say that if the pairs are not in each other's k-nearest-neighbours, then their representations are too separated, which is an 'embedding error' (we use $k = 100$, but the conclusions hold for other 'k' - Supplementary Note 1b). We measure embedding errors on one dataset of hiPSCs[23], and four cell lines from the Human Protein Atlas (HPA)[3]. These are real-world, large, and open datasets of human cells, and they have accurate cell shape segmentations due to cell membrane tagging. We find that prealign-VAE[8,13] has more errors than O2-VAE (Fig. 2f, Chi-Squared $p < 0.001$ for both, hiPSC $\chi^2 = 10.7$, $n = 533$; and HPA $\chi^2 = 17.8$, $n = 1718$). Figure 2e shows image pairs that are errors for prealign-VAE but not O2-VAE: they are pre-aligned globally but are not well-aligned with each other, so they show a failure mode of

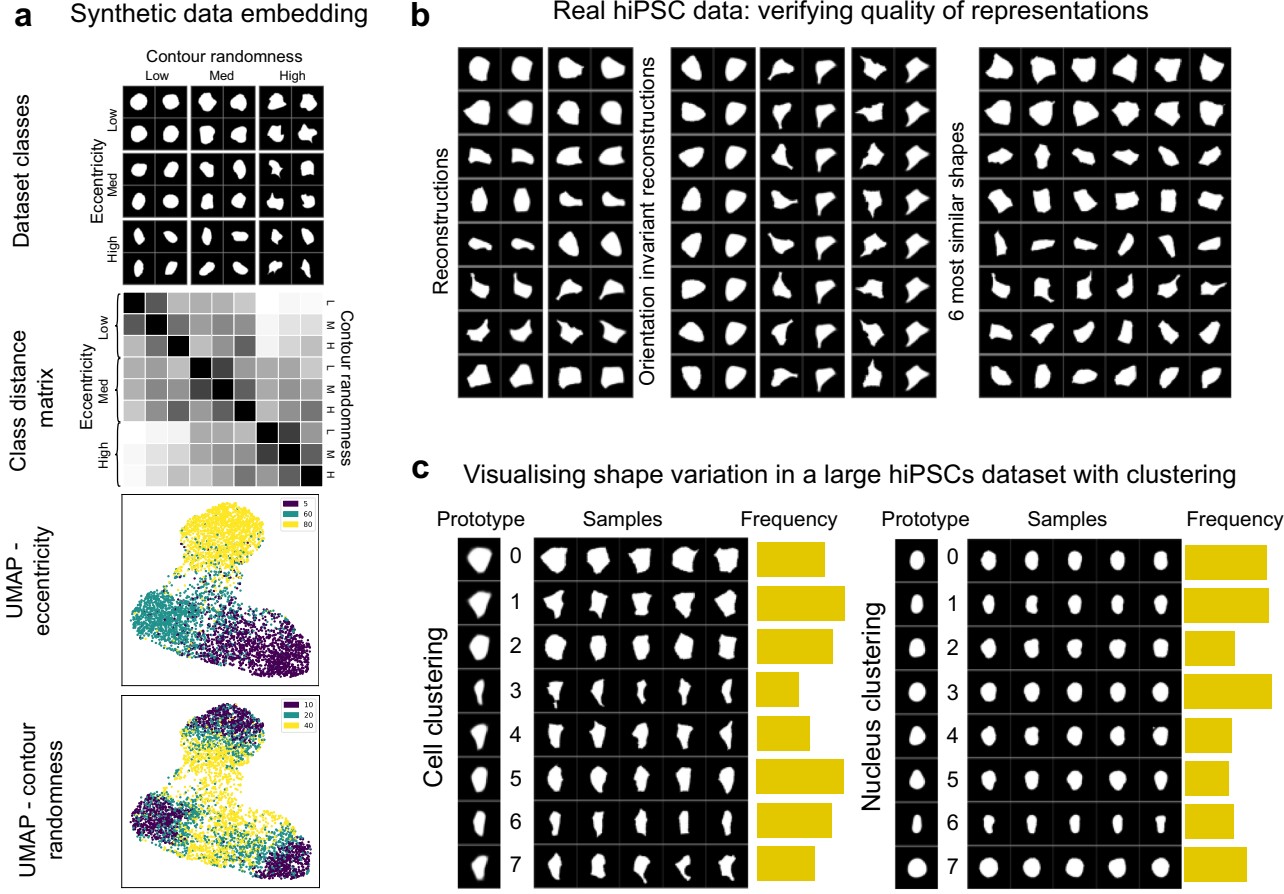

**Fig. 3 | Shape representations in synthetic and real cellular data characterise population variation for exploratory analysis. a** Synthetic dataset and its embedding space: (Top panel) Samples from our synthetic cellular shape dataset with varying eccentricity (columns) and contour randomness (rows), for 9 classes. (Second panel) Distance matrix between robust means of class centroids in embedding space; classes with different eccentricity are further than classes with different contour randomness. (Third panel) UMAP of embedding space coloured by eccentricity and (fourth panel) randomness; these show that eccentricity classes are more separated than contour randomness classes. **b** For real hiPSCs cell shapes without class labels[23], we use model quality evaluation tests. (Left panel) Reconstructions: the original image next to its reconstruction, which should recover the important image features. (Middle panel) Orientation tests: an image in many orientations (left column) should reconstruct images in a canonical orientation (right column). (Right panel) k-nearest neighbours: the first column is sampled images; adjacent images are the `most similar' according to the model's shape space. **c** Still using hiPSCs, we learn a shape space for cells (left) and nucleus (right) segmentation masks, and do GMM clustering with 8 clusters. (Left) `prototypes', reconstructions of the cluster centroid; (middle) samples; (right) cluster frequencies. This communicates shape variation in the data (as well as dimensionality reduction in Supplementary Fig. 12). Source data for (**a**) and (**c**) are provided as a Source Data file.

---

prealignment. The embedding errors lead to errors in downstream analyses like clustering. To show this, we do clustering over the whole dataset and test whether the similar-shape pairs are in the same cluster; if they are not, then it is a 'clustering error'. We repeat this for several clustering methods and hyperparamaeters (Methods) and report the average error. O2-VAE has fewer cluster consistency errors than prealign-VAE (Fig. 2f; Chi-Squared $p = 0.11$ for hIPSC $\chi^2 = 2.6$, $n = 533$; $p < 0.001$ for HPA $\chi^2 = 63.4$, $n = 1718$). In Supplementary Note 1c, we explain the cause of alignment failures, and argue that these issues are likely to persist for any prealignment algorithm. Furthermore, we suggest that prealignment is likely much harder for other greyscale images, and multi-channel images.

### Learning shape representations

One challenge for evaluating unsupervised image-based profiling in cell biology is the paucity of benchmarks with class labels and systematic variation over shape or texture parameters. This was one motivation for releasing the Profiling Cell Shape and Texture (PCST) benchmark that was introduced in the previous section. Sampling shape parameters from a distribution ensures cellular heterogeneity similar to real data where cells in a class have varying eccentricity but

the population-level mean is known. We learn O2-VAE representations of this data, which we dimensionally reduce using UMAP[36] (Fig. 3a). This shows that macro shape is the dominant factor of embedding space variation (Fig. 3a)[36], which is confirmed by PCA reductions (Supplementary Note 2a). Similarly, both the UMAP and the distance matrix between class centroids show that varying eccentricity (macro shape) corresponds to larger distances in embedding space than varying surface randomness (fine-grained shape).

Next, we show that O2-VAE representations can be used for clustering or other downstream tasks. We do clustering with a Gaussian Mixture Model (GMM) with $k = 8$, and report the 'cluster purity', which measures the degree to which a cluster contains instances belonging to only one class. Due to the heterogeneity of the synthetic cells, the cluster purity is 68.6%, which is increased to 87% if considering only mild to moderate cell contour irregularity. Manual review of the classes with high cell randomness show a diverse size and shape that, at times, can resemble multiple different eccentricity or randomness groups (Supplementary Note 2b). GMM has superior purity scores compared with $k$-means (−12%) and agglomerative clustering (−15%).

Next, we model cell segmentations from a real dataset of hiPSCs from the Allen Cell collection[23]. Since there are no class labels, we

propose three tests to verify that the learned representations are meaningful (Fig. 3b). The O2-VAE reconstructs the input images, demonstrating that morphological features are recovered from the vector representation. In the orientation-invariance test, we rotate and flip the image, but the reconstructed image has the same orientation; in Supplementary Note 2cc we quantitatively show that this invariance holds for the whole dataset. In the last panel of Fig. 3b, each row is the 5-nearest neighbours from the first image, demonstrating that small distance in embedding space corresponds to a similar shape.

Large image-based microscopy screens often use methods to rapidly summarise populations and identify prototypical morphology groups for exploratory analysis. We train O2-VAE on the large Allen hiPSC dataset of interphase cells and nuclei[23] with scale-normalisation (Fig. 3c; without normalisation in Supplementary Note 2e) to show the utility of O2-VAE for visualising continuous shape variation. Figure 3c also shows cluster 'prototypes'[37], which are generated by passing the cluster centroid embedding through the decoder. Prototypes for groups with high surface randomness have blurred boundaries. The learned representations allow clustering by shape and rapid quantification of group frequencies, however, cluster boundaries are only approximate for data with continuous variation. GMM clustering with $k = 8$ gives clusters with samples that are similar, but prototypes that are diverse. Cluster frequency charts in Fig. 3c summarise the population shape distribution. We also summarise the dataset with dimensionality reduction approaches in Supplementary Note 2d. Finally, in Supplementary Note 2f, we repeat the clustering for O2-VAE, prealign-VAE, and VAE representations. This shows that VAE representations are worse for clustering: the groupings are confounded by orientation.

## Leveraging shape representations for biological discovery

One application of unsupervised cell profiling is biological discovery and hypothesis-generating research. Here, we show that O2-VAE representations can be used to discover shape subgroups. We model mouse embryonic fibroblasts (MEFs)[7], with two genetic conditions: wild-type ($LMNA^{+/+}$) and lamin-deficient ($LMNA^{-/-}$); and three environmental conditions: 'circular', 'triangular', and control 'not constrained' micropatterns. Images are scale-normalised, trained on O2-VAE, and clustered with GMM, $k = 10$ (Fig. 4a). $LMNA^{+/+}$ cells in all pattern groups have clearly dominant shape clusters (peaks at specific prototypes), but this is lost in $LMNA^{-/-}$ conditions, which have flatter distributions across clusters. This suggests that $LMNA^{+/+}$ has a role in maintaining cell shape consistency. Environmental micropatterning with circles and triangles clearly influences shape: $LMNA^{+/+}$ and, to a lesser extent $LMNA^{-/-}$ are strongly constrained to circular and triangular prototypes respectively. We identify groups with filopodia-like structures branching from thin (cluster 2) or thick (cluster 7) cells. Cells with filopodia-like structures have a lower frequency in $LMNA^{-/-}$ cells, which is consistent with recent studies that suggest $LMNA$ levels may modulate cell shape or filopodia-like projections[38]. We verify previously reported findings[7] that $LMNA^{-/-}$ cells with triangular shapes are more 'blunt', and that nuclei in $LMNA^{-/-}$ cells are more rounded (Supplementary Note 3). A prior work using outline-PCA on this data did not report the groups with changes in filopodia-like structures[7] (see Supplementary Note 3d). In Supplementary Note 3e we repeat the clustering in Fig. 4a using representations from VAE and prealign-VAE. These show that VAE representations tend to erroneously group based on orientation, instead of shape only.

We jointly model cell and nucleus hiPSCs from the Allen dataset[23] to detect mitosis cells which are often filtered in data preprocessing. The UMAP of O2-VAE representations in Fig. 4b shows that mitosis cells are clearly separated from normal cells. Using this approach to label mitosis phases has 89% accuracy, and 99% accuracy for mitosis states other than prophase (Supplementary Note 3). Prior work detected mitosis cells using a 3D classifier with 5000 annotations[23], but our results suggest unsupervised approaches could replace supervised

methods or supplement them by assisting annotation. Next, we measure representation quality using 'linear probing'[39], which is a common evaluation measure[40] of the discriminability of different classes when labels are available (see Methods for details). O2-VAE representations have better linear probing scores (0.7) than prealign-VAE (0.58) and VAE (0.56). Next, we leverage the representation space to suggest plausible shape deformations[8,41] by taking two shapes (Fig. 4c, row edges) and sampling intermediate points in the embedding space (row images). As a final task, we use a GMM model to identify cell and nucleus outliers (Fig. 4b), which are likely bad segmentations or cells in mitosis.

While prior works mostly model cells[5,7–9,13,32,33,42], we show how O2-VAE can learn representations of organelles with complex shapes. First, we train O2-VAE on mitochondria from the Allen hiPSC collection[23] and perform GMM clustering with $k = 14$. In Fig. 4e we show cluster prototypes and the prevalence of each cluster. We observe four superclusters: round puncta (clusters 10–13), small tubes (7–9), larger tubes (5–6), small networks (0–1) and large networks (2–4). The clusters within the puncta and tube superclasses have different thicknesses. We found that large nets were relatively rare, which is consistent with previous literature describing fragmented mitochondria in iPSCs[43]. Next, we explored whether shape groups identified with O2-VAE representations can provide insight into the physiology of mitochondria with various size and shape. Mitochondria interact with multiple other organelles via membrane contact sites. These organelle contacts have different functions: for example, mitochondria-endoplasmic reticulum (ER) contacts are associated with mitochondrial biogenesis and cell proliferation, while mitochondria-lysosome contacts are implicated in mitochondrial remodelling by autophagy[44]. We generated a new dataset of hiPSCs with markers for lysosomes, peroxisomes, Golgi, ER, nucleus, and lipid droplets. Organelles were imaged simultaneously using multispectral imaging, as described in Valm et al.[24]. We measure instances of contact with mitochondria for each labelled organelle (Fig. 4f, g). The dataset is small, so we reuse the O2-VAE model previously trained on the Allen dataset. We extract representations and cluster with GMM, $k = 14$. In Fig. 4g we report perorganelle contact rates, which shows significant variation between mitochondria shape clusters. For example, large nets had particularly high contact rates with lysosomes, though this may be confounded by larger objects having higher random chance of contacting other objects. Therefore, while large nets are rare (Fig. 4e), they may still represent a significant fraction of the mitochondria interacting with lysosomes (Fig. 4g).

## Learning joint shape and texture representations and leveraging them for biological discovery

Although the motivation for O2-VAE is better modelling of shape features, we show preliminary results for the ability to model shape and texture jointly by encoding arbitrary greyscale images. We extend our synthetic cell dataset by adding texture (low, medium, and high) as a third factor of variation (Fig. 5a). First we learn a dataset with constant shape (the same eccentricity and randomness) but variable texture, and the learned representations separate texture classes well (Supplementary Note 4a). Then, we learn an embedding space for a dataset with varying eccentricity and texture in Fig. 5a (nine classes). The UMAP and class distance matrix show that eccentricity is the dominant factor of variation over texture. Extending to the dataset with varying contour randomness (27 classes), the learned embedding space separates the shape classes but only somewhat separates texture; the dominant factors of variation are eccentricity, then randomness, and then texture (Supplementary Note 4b). Figure 5b shows linear probing scores and cluster purity for each factor of variation separately. The representations learned by O2-VAE are consistently more effective at separating texture classes compared with prealign-VAE and VAE. However the scores are much lower than for shape, suggesting there is space to improve texture models.

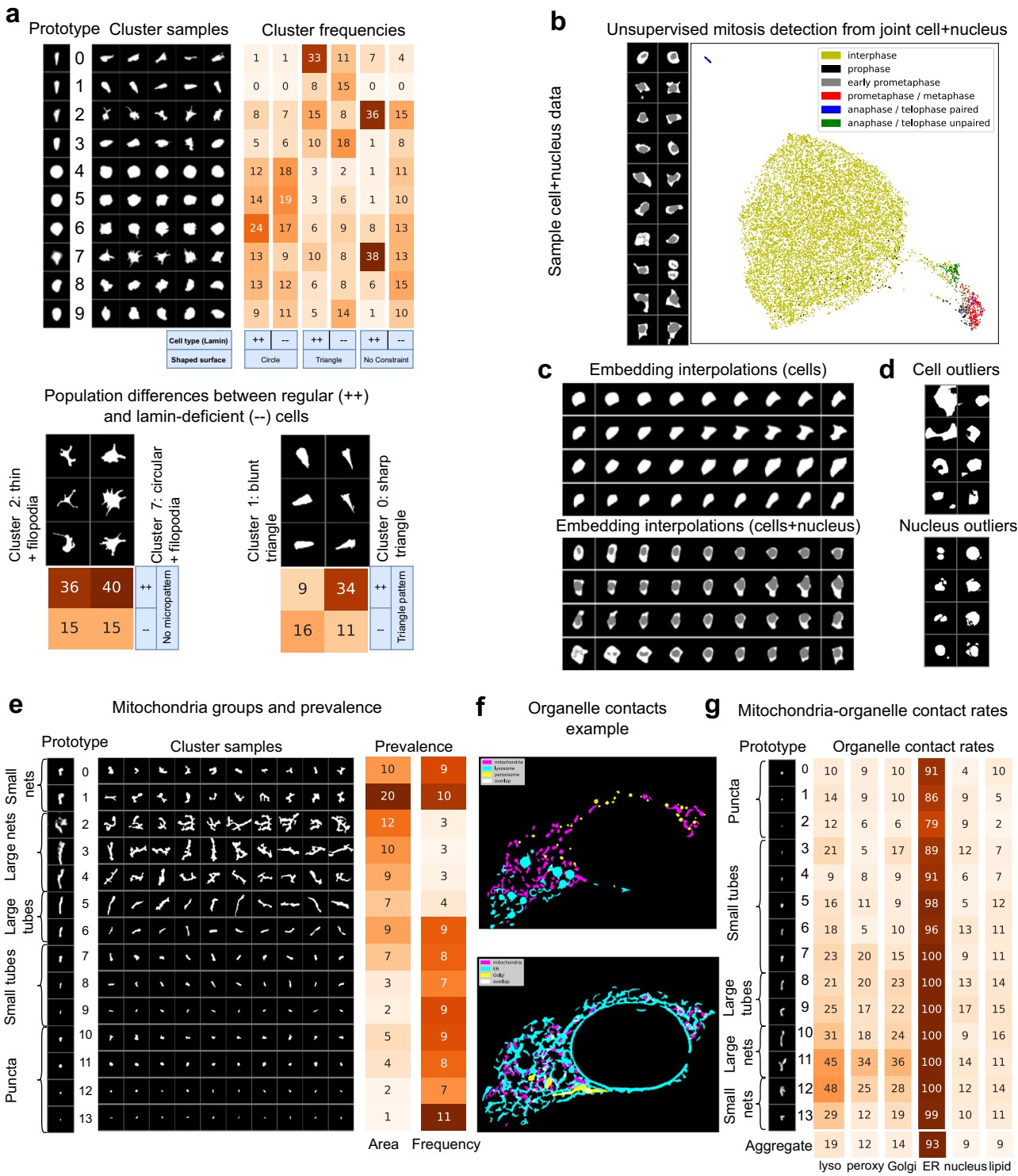

Switching to a real dataset of HeLa Kyoto nuclei undergoing mitosis[25], the classes separate in the embedding space (Fig. 5c). Modelling textural details as well as shape enhances representation quality: O2-VAE representations have higher linear probing scores when trained on greyscale rather than segmentation images (Fig. 5d). Furthermore, O2-VAE has superior linear probing scores than prealign-VAE and VAE.

## Discussion

Autoencoders are a popular method for unsupervised representation learning in biology, which can be a powerful approach for learning features to characterise cell and organelle size and shape, exploratory

image analysis, or biological discovery. We show that autoencoders that fail to enforce orientation invariance have suboptimal representations for clustering and this can introduce errors in downstream analysis. We leverage advances in geometric deep learning to develop a framework for enforcing orientation-invariant autoencoders and introduce O2-VAE. We characterise the structure of O2-VAE embedding spaces for both shape and show preliminary texture results to demonstrate the robust and meaningful learned representations. We demonstrate their utility for analysing and summarising multiple real-world datasets of cells and organelles. Overall, O2-VAE learns better representations compared to standard autoencoders for applications

**Fig. 4 | O2-VAE representations identify meaningful biological subgroups of cells and nuclei. a** We train O2-VAE on mouse embryonic fibroblasts[7] cultured on three micropatterned substrates: circular, triangular, or control (no micropattern); and two treatment groups: wild-type (*LMNA*[+/+]) and lamin-deficient (*LMNA*[−/−]). (Top chart) GMM clustering with $k = 10$: (top, left) prototypes of cluster centroid reconstructions; (top, middle) cluster samples; (top, right) heatmap of relative cluster frequencies per group. (Bottom) highlighting two group differences. Loss of LMNA (−/−) correlates with reduced filopodia-like structures, a novel finding not identified by prior methods to the best of our knowledge. Low LMNA (−/−) groups have lower prevalence of triangular classes that have sharp edges. **b** (Left) example cell+nucleus multistructure hiPSCs[23], and (right) UMAP of embedding space coloured by mitosis class. Mitosis cells separate from normal cells: this is an unsupervised detection method. **c** Interpolations of cell and multistructure hiPSCs are candidate shape deformations. Images at row edges are real cells. We sample

points between their embeddings and reconstruct them to form the other images. **d** Fitting a GMM to learned representations enables detection of outliers for cells and nuclei in hiPSCs; they are likely bad segmentations or in mitosis and can be filtered in preprocessing. **e** Mitochondria in Allen Cell collection data clustered with GMM, $k = 14$. (Left) cluster prototypes, (middle) samples, and (right) prevalence, where `area' is the percent of mitochondrial area in that cluster and `frequency' is the percentage count of mitochondria objects in that cluster. **f** Example cell with pseudo-coloured segmentation masks with other organelles, that have many contacts (overlapping organelle pixels in white). **g** For hiPSC data, clustering with GMM and $k = 14$ (cluster samples and prevalence in Supplementary Fig. 27). Contact rates of each mitochondrial shape group with each organelle, which is the percentage of mitochondria in the group in contact with that organelle. `Aggregate' is the contact rate over all clusters. Supplementary Note 3c shows results per sub-experiment.

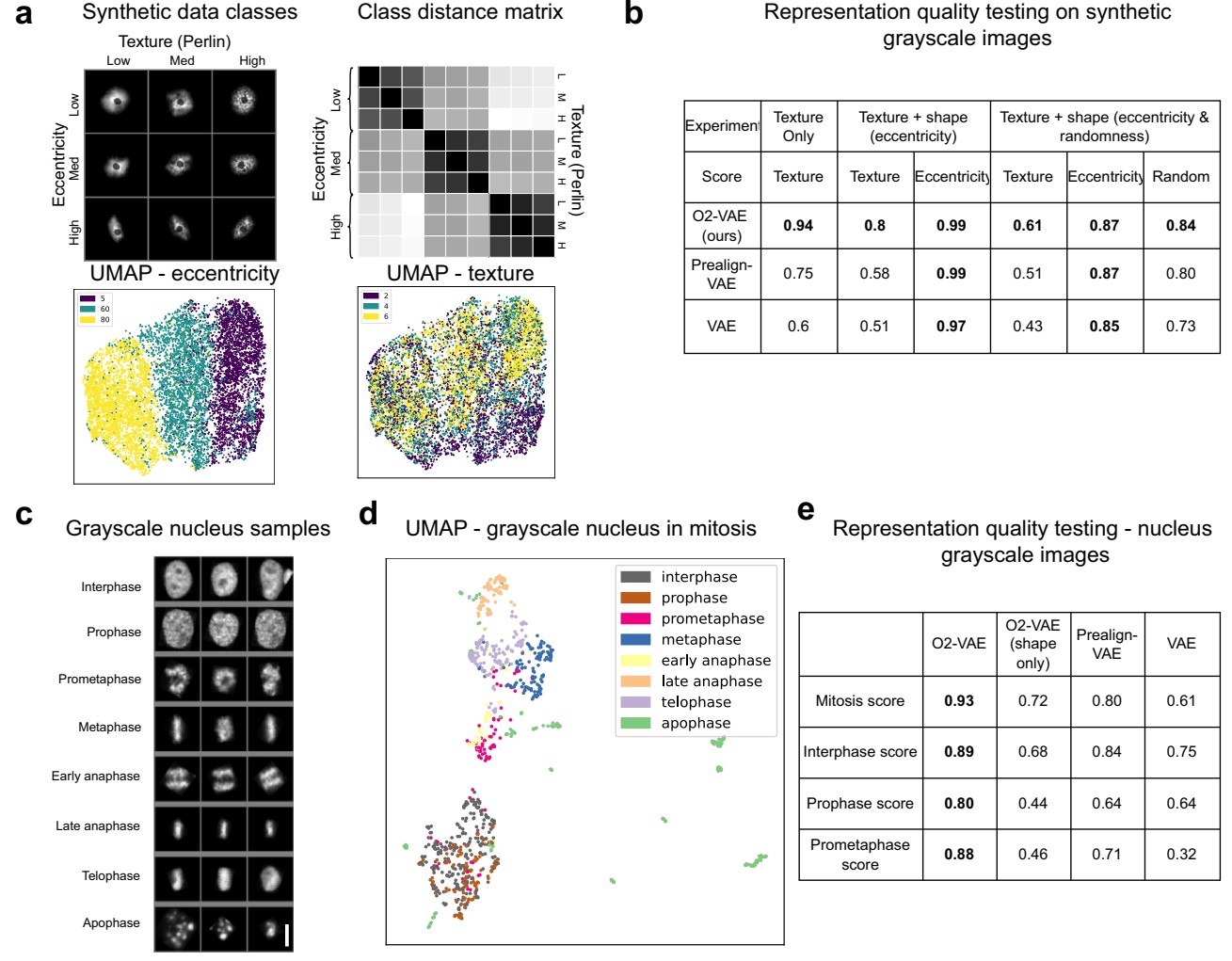

**Fig. 5 | Modelling greyscale images enables joint representations of shape and texture. a** Synthetic dataset and its embedding space: (Top) Samples from our synthetic cellular shape and texture dataset with varying eccentricity (columns) and Perlin texture (rows), for 9 classes (no scale bar because data is synthetic). (Second panel) Distance matrix between robust means of class centroids in embedding space; classes with different eccentricity are more separated than classes with different texture. (Third panel) UMAP of embedding space coloured by eccentricity and (fourth panel) texture; these show that eccentricity classes are more separated than texture classes. **b** Linear probing scores measure representation quality by simulating a classification task on the embedding space (top scores in bold). For three experiments modelling texture or texture and shape

jointly, O2-VAE has better texture representation scores (see Methods). **c** Sample data of real nuclei with mitosis phase classes after min-max scaling[25]. Scale bar in bottom-right image, 10 μm. **d** UMAP of representations have good separation of classes, with some mixing of interphase and prophase cells. **e** Representation quality (linear probing) scores for all mitosis classes and the three challenging classes (interphase, prophase, prometaphase)(top scores in bold). O2-VAE representations perform better than VAE baselines. O2-VAE representations are better when modelling texture and shape jointly (using greyscale images) compared with shape only (using segmentation images). Source data for (**a**) are provided as a Source Data file.

where evaluation labels are available. Since there are few labelled datasets for representation evaluation, we generate and release a synthetic dataset with varying shape and texture.

Our work extends and complements recent representation learning models based on autoencoders in the unsupervised[13,14] and self-supervised[15,16,22] settings. Since we only constrain the encoder representations and we automatically re-align the reconstruction, O2-VAE can be combined with other methods that change the architecture or loss. For example, this approach could be combined with the recent self-supervised method Cytoself[15]. Our analysis of Cytoself in Supplementary Note 5b shows that orientation can be a confounder for grouping images based on protein distribution in this model. Our approach can also be extended to enforce other invariances, including translation[30], axial rotation[45], and scale[29]. It can be extended to model 3d images[30,45]. We experimented on centred objects, but O2-VAE is extensible to model complex biological images including protein localisation micrographs or electron micrographs.

Although the challenges of prealignment was one motivation for developing O2-VAE, we also hypothesised that orientation invariance improves autoencoder representation quality more generally. We have empirical evidence to support this hypothesis: on the simulated datasets, O2-VAE has better linear probing scores of texture and contour randomness; the margins are high and not well explained by prealign errors. Further, prior work shows that orientation-invariant encoders have better performance for *supervised* models on biological and medical data[46–48], despite supervised learning having more techniques to enforce invariance (e.g. data augmentation). There are also theoretical arguments supporting this idea: a central idea in geometric deep learning is that enforcing known data invariances in neural networks should improve representations[26,49]. Having said that, the importance of explicitly modelling orientation invariance in supervised learning may depend on the problem formulation, dataset, and specific implementation (e.g., network architecture, loss function).

Although our work focuses on autoencoder models and shape features, we acknowledge that other profiling methods are used for cell profiling, especially orientation-invariant engineered features[50,51] and contrastively learned features[52,53]. In Supplementary Note 5a, we quantitatively compare O2-VAE to these alternatives for all datasets that have ground-truth class labels. For profiling shape features, O2-VAE is competitive with all baselines and superior to autoencoder baselines. For profiling texture features our O2-VAE performs better than baseline autoencoders. However, it performs worse than engineered or contrastively learned features. We hypothesise that the gap is due to O2-VAE implementing the convolutional-only architecture with standard VAE loss[11], which is known to poorly generate texture details[11,54]. Machine learning literature shows that modifications to the VAE can improve texture modelling, for example with the VQ-VAE architecture and loss function[55,56]. This updated architecture has been shown to model texture profiles well in fluorescence microscopy[15]. Even if autoencoder models do not beat alternatives on profiling benchmarks, they have advantages in that they can generate data from their latent space, which can aid in interpretability and discovery[14,41]. Future work can extend O2-VAE to include new architectures, enforce additional invariances, and explore 3D implementations.

We also acknowledge that O2-VAE may not be appropriate for all use cases. Prealignment can sometimes work well, for example where there are meaningful landmarks for alignment[5]. Some analyses may draw insights despite imperfect prealignment, in which case major-axis alignment - which aligns objects only up to flips - may be sufficient or preferred for simplicity[23]. For basic shapes, simple metrics like area and perimeter may be adequate. For datasets where the interesting factor of variation is not shape, but is texture or particle colocalisation, other methods may be preferred: especially engineered features[50,57] or contrastive methods[52]. Although it may be difficult to train O2-VAE with very small datasets, models pre-trained on large datasets can transfer to similar but smaller datasets, for example, we transferred mitochondria representations from the Allen collection to our multi-organelle dataset[23]. Finally, while our method requires some familiarity with deep learning, we provide example Python notebooks for using O2-VAE.

Computer vision methods developed for natural images have driven many recent advances in bioimaging, but these approaches do not leverage the inherent structure and symmetry in biological images. We show that by taking advantage of these properties and encoding orientation invariance into O2-VAE, we improve the learned shape representations and subsequent performance of autoencoders on downstream tasks. We believe that existing and future profiling methods may benefit by encoding orientation invariance. More generally, we hope that our work stimulates interest in the question of how algorithms from the machine learning community can be adapted and improved for biology applications by incorporating expert domain knowledge and the intrinsic properties of the data.

## Methods

### Ethical Statement
All research complies with relevant ethical and institutional guidelines.

### PCST: Profiling Cell Shape and Texture (PCST) benchmark
Synthetic cells were generated using a modified version of SimuCell version 1.0[35] in MATLAB R2020a following the guidelines in the SimuCell example scripts. For each image, one cell was centred on a black background of a $512 \times 512 \times 3$ image. Cells were synthesised using parameterised distributions of cell/nucleus radius, eccentricity, cell shape contour irregularities (aka cell randomness), base fluorescence intensity per object, and texture. All parameters were fixed except for those systematically varied, which included cell eccentricity (0.05, 0.6, 0.8), cell randomness (0.1, 0.2, 0.4), and Perlin texture length scale (2, 4, 6). All combinations of cell eccentricity and cell randomness produce 9 unique shape groups. When texture is incorporated, there are 27 unique conditions representing all combinations of shape and texture parameters. A total of 15,000 images were generated for each condition. Each image was assigned a unique random seed, which was set before and during synthesis to ensure shape and texture were controlled across conditions. This created "triples" or images across three conditions which have two parameters constant and only one parameter varied. For example, conditions 1, 2, and 3 have constant eccentricity 0.5 and cell randomness 0.1 with variable Perlin length scale. The same seed was set for image 1 across conditions 1, 2, and 3 and thus the cell will have identical size and cell contour irregularities with variable texture. Additional information on the interpretation of the parameters as well as detailed description of all settings can be found in the source code and dataset.

### Multi-organelle iPSC data: cells and culture conditions
Previously generated human iPSCs (KOLF2.1J) inducibly expressing neurogenin2[58] were incubated at 37 °C, 5% $CO_2$ and cultured under feeder-free conditions in StemFlex medium (Gibco A3349401) on Vitronectin (VTN-N Recombinant Human Protein, Truncated, Gibco A14700). The cells were passaged using ReLeSR enzyme-free stem cell selection and passaging medium in the absence of Rho-associate kinase (ROCK) inhibitor.

### Multi-organelle iPSC data: seeding, transfection, and labelling
The cells were split as colonies using ReLeSR and seeded in chambered coverglass for high-resolution microscopy (Thermo Scientific, Nunc™ Lab-Tek™ II Chambered Coverglass #1.5, 155379). Transfection was done one-day post-splitting. Approximately 20 min before transfection, media was replaced with Essential 8 (Gibco, A1517001). The transfection mix was prepared in Opti-MEM 1X (Gibco 31985070) with 0.5 µl of Lipofectamine Stem Reagent (Invitrogen, STEM00001), and 500 ng of total DNA divided as follows: 167 ng of pEIF1a::Transposase (gifted by Dr. Michael Ward), and 333 ng of organelle markers:

lysosomes [pEIF1a::LAMP1::mTurquoise], mitochondria [pEIF1a::Cox8(1-26)::eGFP], Golgi [pEIF1a::Sit::OxVenus], peroxisomes [pEIF1a::mOrange2::SKL], endoplasmic reticulum [pEIF1a::Sec61$\beta$::mApple] (Twist Technologies). The mix was applied dropwise and cells were incubated for four hours before the media was changed back to StemFlex and supplemented with BODIPY™ 665/676 (80 ng) for the staining of lipids (Invitrogen, B3932). Prior to imaging, a media change was performed.

## Multi-organelle iPSC data: live microscopy

Images were acquired on a Zeiss 880 laser scanning confocal microscope equipped with a 32-channel multi-anode spectral detector (Carl Zeiss) in lambda mode at 8.9$nm$ bins (collecting wavelengths 410–695), with 63 × /1.4 NA objective lens, and a 2.2 × zoom. All fluorophores were excited simultaneously using 405, 458, 514, 594, and 633 nm lasers, with a 458/514/561/633 nm main beam splitter. Z-stacks were acquired with 50% overlapping 1.2 µm slices, at a scan speed of 1.90s/frame. Live imaging was performed with stable 5 $CO_2$ at 37 ℃, and each imaging session was not longer than 1–2 h. Images of multiply labelled cells were subjected to linear unmixing using Zen Software (Carl Zeiss) by using single fluorophore reference spectra as previously reported[24].

## Data processing: public datasets

The Allen Cell hiPSCs dataset[23] was downloaded from here, which has cell and nucleus segmentations, and mitosis annotations as described in refs. 23,59. We crop the 3D cell area, and choose the middle z-slice as the 2d cell representation. For Allen mitochondria datasets, we use this same z-slice. We sample 10,000 cells, and when training O2-VAE models, we randomly assigned 20% to the held-out validation set. For mitochondria, we sample segmentations from the same z-slice, and follow the same sampling procedure.

For Human Protein Atlas data[3], immunofluorescence images and cell segmentation masks were downloaded from https://www.proteinatlas.org/about/download[3]. We used the following cell lines: A549, A431, SK-MEL-30, and SiHa.

The mouse embryonic fibroblast (MEFs) dataset was generated and described in[7]. In brief, MEFs were cultured on circle or triangular fibronectin micropattern surfaces to enforce cell shape constraints versus control, non-patterned surfaces. The cells were stained with Phalloidin to label actin and DAPI to label nuclei. We segmented cells from immunofluorescence images of Phalloidin-labelled cells to obtain a cell segmentation mask using a custom pipeline combining an ilastik v1.3.3[60] pixel classifier with CellProfiler v4.2[51]. The segmentation results were reviewed by a cell biologist to ensure accuracy.

The HeLa Kyoto nuclei images are from the CellCognition project[25]. We use the H2B-RFP tagged channel that is provided already segmented and cropped, and we use the provided train/validation splits.

For all cells images, the object was centred using the arithmetic mean of pixels, then cropped to 512 pixels to cover the largest objects, and resized to have side pixel length of 64, 128, or 256. Multi-channel cell and nucleus objects were centred based on the cell shape. For datasets with 'scale-normalisation', the scale measure is `axis_major_length` property from the `regionprops` function in scikit-image[61], and we resize objects using the `torchvision`[62] resize function.

## Data processing: segmentation of fluorescent multi-channel organelle dataset

Organelle channels are segmented using the Allen Cell and Structure Segmenter[59]. We use the 'classic' filter-based workflow starting with the recommended parameters, and updating them based on visual inspection.

## Model component: orientation-invariant encoder

The orientation-invariant encoder is constructed similarly to the models presented in ref. 19. We use O(2)-equivariant steerable CNN layers[63]. Briefly, steerable CNN's are similar to standard CNN's, but where the convolutional filter weights are defined as the sum of 2d basis functions (and the learned parameters are the coefficients for those basis functions). By choosing 2d basis functions that are O(2)-equivariant, the final convolutional filter is also O(2)-equivariant. We use the implementation of steerable CNN's from the e2cnn library[19].

We stack six equivariant blocks, where each block has an equivariant convolutional layer, a batch normalisation layer,[64] and gated ELU nonlinearity[65]. This block was proposed in ref. 30 and recommended by ref. 19. Next, we then spatially pool (average) each feature channel to dimension 1: aggregating spatially over an equivariant function makes that function invariant. We reshape the input to a vector, and pass it through one fully connected layer. The output vector has size $2 \times d$, where $d$ is representation dimension. Note that due to discretisation and feature downsampling, it is not possible to guarantee perfect invariance in O(2), which is why we validate that the learned model really is invariant (see Methods - Representation Quality Tests).

## Model component: realigning reconstruction with the input

Reconstructed images are misaligned with the input because the encoder destroys orientation information. We rotate and reflect the input image - called image registration - to best align with the reconstruction (Fig. 1a). This is done efficiently using Fourier space methods[20], and we provide an implementation in PyTorch[62] to register a batch of images in parallel on a GPU (see https://github.com/jmhb0/). For both images, we transform to polar coordinates, take their Fourier transforms, and then compute their cross-correlation. The point of maximum activation of the cross-correlogram in polar coordinates corresponds to the re-alignment angle. We perform the same process on the reflection of the first image, and if the resulting correlogram has higher correlation, we use the reflected input image, otherwise we choose the non-reflected image. The additional overhead from computing image alignment in our experiments does not exceed 3% of time to compute one forward and backward pass of O2-VAE training (where forward and backward passes take up at least 90% of training time; these tests were for images with dimension 64 and 128).

A simpler re-alignment algorithm can also be used: choose an angle increment (e.g., $\theta = 1$°) and create $360/\theta$ rotated copies of the input image. Then simply measure the correlation of each rotated copy and choose the one with highest correlation. This is simpler and possibly less prone to numerical errors, but it consumes much more memory (that scales with the square of image width), and therefore requires very small batch sizes and slower training.

## The O2-VAE model

The O2-VAE follows the variational autoencoder framework[11,12] (Fig. 1a). The image is passed through the orientation-invariant encoder (described above) to form a vector bottleneck layer with length $2 \times d$ (where $d$ is a dimension hyperparameter). As in the original VAE[11], this vector is the parameters of a Gaussian distribution with diagonal covariance. The $d$-dimensional 'mean vector' is the 'image representation'. We sample a vector from this Gaussian and pass it through a (not-orientation-invariant) deconvolutional decoder[27] with 5, 6, or 7 blocks for image sizes 64, 128, and 256 respectively. Each block has a transposed convolution layer, a batch norm layer and an ELU nonlinearity. The decoder output is the reconstruction. We re-align the input image to align with the reconstruction (described above).

In training, we apply a reconstruction loss that penalises pixelwise distance to the input and reconstruction; we use binary cross entropy for segmentations and mean square error for greyscale images. We also use the VAE distribution loss that penalises the KL divergence between the bottleneck distribution and an isotropic Gaussian[11], and we weight this loss term by $\beta$. We use $\beta = 0.01$ and embedding dimensions $d = 128$ for all models trained in this paper. We

use the Adam optimiser[66] with learning rate $10^{-3}$, which we train until the loss on the held-out validation set converges.

## Testing orientation sensitivity with synthetic data

For tests in Fig. 2a, the ellipses dataset is generated with 1000 images each in classes 0 and 1. Class 0 objects has minor axis radius of 10 and a major-axis radius sampled uniformly between 15 and 17. Class 1 objects have minor axis radius 10 and major-axis radius uniformly sampled between 23 and 25. The objects are centred in a $62 \times 62$ pixel image, and randomly rotated. Half the data is assigned to the train and half to a held-out validation set. All analyses (figures and clustering results) are done on the test set. The trained model is the same as the O2-VAE (described above), except the encoder is a 6-layer *****(non-invariant) convolutional encoder, and we do not re-align the input with the reconstruction. We train with the more standard $\beta = 1$ for 200 epoch when the validation loss converged. We use only a $d = 2$ representation to allow direct visualisation of the representation space. We also test embedding dimensions, {4,16,32,128}; in higher dimensions the results are inconsistent: $k$-means clustering scores either below 55% or 100%, and the results are unstable between different runs of the same model with different random seed.

## Prealignment algorithms

Prior works do prealignment in a one or two-stage process[5,8] (Supplementary Fig. 1). There is variation in the details, but we use the following procedure. In the first stage, the 'major axis' is computed and aligned to the y-axis. One definition of the 'major axis' is the major axis of the first-order ellipse that has the same second moments as the binary image, which we compute using scikit-image[61]. An alternative definition is the first primary component from a PCA, where the input data are the flattened x-y coordinates of either the image pixels or of evenly-spaced points along the shape outline. Both approaches require a segmentation to compute the angle.

The major-axis alignment only determines orientation up to flips in the x or y-axis, so a second stage updates the final orientation. The approach is to register each cell against a single reference. For some cells, the objects may have a 'natural' orientation (e.g., the outline of *Drosophila*[5]) which can be used as a reference. But when no orientation is known, a common choice is the mean cell over all images. For each image, we choose the flip that minimises the image cross-correlation with the mean cell. After choosing the flip for each image, the mean cell will be different, so this process is repeated iteratively until the average cross-correlation loss converges, which took fewer than 30 iterations for our datasets.

## Quantifying embedding errors

We first identify 'high-confidence similar object pairs'. Since we do not have a measure for semantic similarity, we use a pixel-based measure as a proxy. For centred images $(x, y)$ with image dimensions $(m, n)$, their cross-correlation (or root mean squared error) is:

$$RMSE(x,y) = \sqrt{\left( \frac{1}{m \cdot n} \sum_{i,j} (x_{i,j} - y_{i,j})^2 \right)} \quad (1)$$

This score will be larger for larger shapes: if one edge of the cell is perturbed, then more pixels will be different between image pairs if the object is larger. We therefore normalise it with $C = \sqrt{\max\{s(x), s(y)\}}$, where $s(\cdot)$ is the number of object pixels, (which we justify further in Supplementary Note 1d). This is the normalised root mean square error, or *NRMSE*. We register $x$ and $y$ by finding the rotation and flip of $y$, called $\hat{y}$, that minimises the *NRMSE*, which we compute using the same procedure described in 'Realigning reconstruction with the input'. We define the set of high-confidence pairs as those having *NRMSE* below a threshold. By choosing a conservative threshold, this approach for identifying semantically similar pairs has low false

positive rate, but since it is a proxy measure, the false negative rate may be high. This is the best result we can achieve without ground-truth labels for semantic similarity. We choose a range of thresholds for the clustering experiment, and we show sample pairs in Supplementary Fig. 6 to verify that threshold is reasonable.

To measure 'embedding errors', we label the nearest neighbours to each image. For image pair $(x_i, y_i)$, the neighbour separation score $k_i$ means either $x_i$ is the $k_i$th nearest neighbour to $y_i$ or vice versa (whichever is smaller). For a threshold, $t$, a 'high-confidence similar pair' has an embedding errors if $k_i > t$. We choose $t = 100$ for experiments and plot this for many $t$ in Supplementary Fig. 2, which shows the conclusions are consistent across $t$. This order-based metric of similarity is preferable to a distance-based metric because we want to compare different representation spaces, and the notion of distance in one space does not transfer to another. We repeat this for O2-VAE and prealign-VAE. Next we measure 'clustering errors', where a 'high-confidence similar pair' is assigned a different cluster. Since clustering is sensitive to hyperparameters, we perform many experiments: two clustering methods ($k$-means and Gaussian mixture model), six values of $k$, {10,12,14,16,18,20}, four thresholds for 'high-confidence' NRMSD, and 20 random seeds. We compute clustering error rate for each experiment and then report on the average. This establishes that differences in error rates are general, and not a result specific to hyperparameter choices.

We generate confidence intervals and $p$-values for the results in Fig. 2b, f using functions from SciPy[67]. In these analyses, each data point is a pair of cells that should have similar embeddings or be in the same cluster. Each data point is a binary variable indicating whether it is an 'error'. For confidence intervals, we use the nonparametric bootstrap, from scipy stats package with default parameters. For p-values, we do a Chi-square test, specifically the `chi2_contingency` function. The groups are the choice of embedding space: O2-VAE vs prealign-VAE. The categories are 'is error' and 'not error'.

## Representation space quality tests

The verification tests in Fig. 3b do not require access to labels, so they are useful in real analyses to check that the embedding space is meaningful. The reconstruction test simply samples an image, computes the reconstruction, and re-aligns the objects (described above). The kNN test sample images and displays the images with nearest Euclidean distance in embedding space. Both these tests require the judgement of an experimenter to asses whether important features are present (in reconstructions), and whether pairs of objects really are similar (in kNN). Since they require manual inspection, we recommend sampling uniformly across the data.

The qualitative orientation test samples an image, makes rotated and reflected copies, and outputs the image reconstruction without re-alignment. A properly orientation-invariant encoder destroys orientation information, so we expect that non-aligned reconstructions are all the same. A second quantitative test (in Supplementary Note 2h) is more comprehensive and directly tests the representations themselves. We rotate the object in increments of $\theta = 20$ on the original and reflected images, then measure the Euclidean distance between these embeddings at each orientation. Since discretisation effects cause some representation difference, they will not have identical embeddings, but they should be very close. The test is passed if the biggest embedding distance between different orientations is less than the embedding distance to every other image in the dataset. In practice this test may fail by a very small distance margin if the data have duplicates or near-duplicates, in which case we recommend permitting a small distance margin. We observed this problem for MEFs data that had collected the same cell in different images, and for mitochondria data where some small round puncta had identical segmentation masks. We found that this quantitative test passed in trained O2-VAE models, but did not always pass in untrained models. This suggests that the orientation invariance, which can only be approximately

enforced by the architecture, is strengthened by the training procedure, especially through the orientation-invariant reconstruction loss.

## Clustering, dimensionality reduction, and outlier detection

We perform GMM and $k$-means clustering of representations in scikit-learn[68]. We do clustering for two cases: where the ground-truth classes are known or unknown. For most experiments (those with real data), where the ground-truth classes are unknown, we determine $k$ experimentally. There are two factors we weigh for choosing $k$: small $k$ leads to groups that have too much intra-class variation; too high $k$ leads to too many groups, which makes it harder to interpret analyses over those clusters like the frequency charts in Fig. 3d. We tried a range of values for $k$ and visualised sampled shapes from those classes. We chose a $k$ where the within-group variance was low. In some cases, like Fig. 4e, this led to some groups having similar properties, so we defined 'superclusters' that combined clustering groups, and we visualised them next to each other in results tables to make interpretation easier (Fig. 4e). The superclusters are chosen by visually comparing shape similarity of the prototypes; this can also be informed by measuring the distance between cluster centroids. Ultimately, the results rely on the judgement of the experimenter. For example, this required $k = 14$ for mitochondria, but only $k = 10$ for MEFs. Because computation complexity for fitting GMMs scales cubically with dimension, $d$, we use PCA to reduce to 32 dimensions, which retains at least 90% of explained variance for all our experiments. For figures, we reorder clusters so that similar shapes are near to each other. We implement this by agglomerative clustering over cluster centroids with scikit-learn[68], and then by taking the ordering of the resulting dendrogram. This aids interpretability, for example, in Fig. 4e, the mitochondria superclusters have their subclusters ordered together, though this procedure does not always put similar-shape groups close (which is not possible in the general case for a 1d ordering). For displaying cluster samples, we perform multiple approaches. The easiest approach is uniform sampling. Another approach is to get a score for 'cluster confidence', and display a spread of 'low' and 'high' confidence samples. The 'cluster confidence' for GMM clustering is the probability score, and for $k$-means it is the negative of the clustering objective. By showing a range of low- and high-confidence objects, we can asses cluster coherence. For example, this revealed the $k$-means clustering of mitochondria latent spaces tended to assign large networks to many clusters (with low confidence) that had very different shapes.

For experiments with synthetic datasets, PCST, we have ground-truth class labels. When testing clustering, we use the ground-truth number of clusters as $k$. We report on 'cluster purity' with the following definition. Do clustering; assign each cluster to only one ground-truth class using Hungarian matching[69]; the cluster purity of each cluster is the percent of data in that cluster assigned the correct class; take the average of the purity scores for each cluster.

For PCA axis traversals, we perform PCA over image embeddings with scikit-learn[68]. Row $i$ is formed by sampling the data centre, and then sampling embedding space points in the range $[-2\sigma, 2\sigma]$ along that PC, where $\sigma$ is the standard deviation of data projected to that PC. UMAP reductions are computed using the official Python implementation[36]. Visualisations of shape in the 2d reduced space (Supplementary Fig. 12) is done by sampling grid points in the 2d plane, and finding the nearest real image in the dataset.

For outlier detection (Fig. 4d), we take a probabilistic approach[70] by fitting a 20-component GMM to representations, and using the probability under the model as an outlier score. For mitosis detection (Fig. 4b), the outliers are determined simply by manual inspection of the UMAP reduced data to 2d. In our experiments, varying UMAP hyperparameters gave consistent predictions.

## Testing representation quality with linear probing

Linear probing is a very common technique for evaluating representation quality for deep learning models[39]. We randomly split data into 80% train and 20% test sets, and then fit a logistic regression classifier with L2 regularisation to the train set using scikit-learn[68]. We evaluate the classifier on the test data; we report the F1 score for each class, and the macro-average F1 score across classes. In Results, we use linear probing in two ways. For datasets with labels for ground-truth classes (groups for simulated data; multistructure cell and nucleus in mitosis; greyscale in mitosis) we probe using these labels. Where there are labels for each generative factor (eccentricity, contour randomness, Perlin texture), we perform separate tests where data is labelled by only one generative factor at a time. This allows us to measure, for example, the separability of representations based on eccentricity independently from texture.

## Organelle prevalence and contact frequencies

In Fig. 4e, we report cluster prevalence by summing the total area from all mitochondria in that cluster ('area') and then counting the mitochondria ('frequency'). Then for Fig. 4g, we take all pairs of organelles, for example, all mitohconria and all lysosomes in a cell, and measure the minimum distance between pixels in those objects. An object is 'in contact' with another object if its separation is smaller than 2 pixels (0.160 μm). For example the contact frequency of mitochondria with lysosome in cluster 1 is the total number of cluster 1 mitochondria that contact a lysosome divided by the total number of cluster 1 mitochondria. The conclusions reported in Results were consistent over threshold ranges from 0 to 4.

## Statistics and Reproducibility

A two-sided Chi-square was performed for comparisons in Fig. 2f using the SciPy `chi2_contingency` function. The $\chi^2$ test statistic is given. An exact $p$-value is given except when $p < 0.001$. The exact sample sizes are given when statistical tests are performed on groups. Error bars are 95% confidence intervals (CI) computed via nonparametric bootstrap using SciPy `stats.bootstrap` function with 1000 resamplings and default settings. No data were excluded from this analyses.

## Reporting summary

Further information on research design is available in the Nature Portfolio Reporting Summary linked to this article.

## Data availability

Data generated in this study: All data generated in this study are publicly available, as described below. The synthetic dataset Profiling Cell Shape and Texture (PCST) generated in this study has synthetic fluorescence cells created by sampling from parameterised distributions with a known population mean eccentricity, contour randomness, and Perlin texture. The data is publicly available under a CC BY-SA 4.0 at https://zenodo.org/record/7388245#.Y4k1OezMJqs. Synthetic cell shapes and textures were generated using a modified version of SimuCell v1.0[35] (https://github.com/AltschulerWu-Lab/simucell)in MATLAB R2020a. The multi-organelle iPSC dataset generated in this study has live undifferentiated KOLF2.1J human iPSCs labelled with 7 organelle markers targeted to the nucleus, endoplasmic reticulum (ER), Golgi, mitochondria, peroxisomes, lysosomes, and lipid droplets. The data are publicly available under a CC BY 4.0 license at the Bioimage Archive with the accession S-BIAD71 at this url https://www.ebi.ac.uk/biostudies/bioimages/studies/S-BIAD712. For the plots in the figures, Source Data are provided with this paper. Pubic datasets: Public datasets were used in accordance with the dataset license and their information is shown as follows: Allen Cell[23]: Fluorescently tagged hiPSC lines that target many key cellular structures and substructures. Includes cell/nucleus segmentations and mitosis annotations. https://open.quiltdata.com/b/allencell/packages/aics/hipsc_single_cell_image_dataset and the DOI reference for the paper https://doi.org/10.1038/s41586-022-05563-7. Human Protein Atlas (HPA)[3]: The HPA subcellular atlas shows the fluorescence expression and distribution of proteins encoded by >13,000 genes using numerous different cell lines and antibody-based

immunocytochemistry. https://www.proteinatlas.org/about/download. Mouse embryonic fibroblasts (MEFs)[7]: MEFs with or without Lamin A knockout stained with DAPI (nucleus) and Phallodin (actin) and seeded on Circle and Triangle micropatterns coated with Fibronectin. https://github.com/kukionfr/Micropattern_MEF_LMNA_Image. HeLa Kyoto CellCognition[25]: Human HeLa Kyoto cells expressing fluorescent markers of chromatin histone-2B and alpha-tubulin. https://cellcognition-project.org/demo_data.html.

## Code availability

We release our code for use by the community at https://github.com/jmhb0/o2vae/, or at https://doi.org/10.5281/zenodo.10206848[71]. All data analysis was performed using Python 3.9. A list of Python package dependencies can be found in the `requirements.txt` file of the GitHub repository. This repository contains documentation and example Python notebooks for training models and analysing representations.

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

## Acknowledgements

We thank our colleagues Kuan-Chieh Wang and Aditya Grover for insightful discussions and feedback. We thank Bill Skarnes for providing KOLF2.1 cells. We acknowledge support from the following sources: a Chan Zuckerberg Initiative Neurodegeneration Collaborative Pairs Award (S.Y.L. and S.C., 2021-235009); a Stanford Alzheimer Disease Research Center NIH grant #P30AG066515 (J.J.N.); a National Institute of General Medical Sciences grant R35GM133460 (S.C. and M.C.Z.); the Arc Institute (A.L.). S.Y.L. is a Chan Zuckerberg Biohub - San Francisco Investigator.

## Author contributions

JB: Conceptualisation, Formal analysis, Methodology, Software, Visualisation, Writing - Original Draft, Writing - Review & Editing. JJN: Data Generation, Software, Supervision, Validation, Writing - Review & Editing. MCZ: Data Generation and Curation. AL: Software. SC: Data Curation, Supervision, Resources, Writing - Review & Editing. SYL: Conceptualisation, Supervision, Resources, Writing - Review & Editing. All authors reviewed and provided feedback on the manuscript.

## Competing interests

The authors declare no competing interests.
