## [Peer Review File · Nature Communications]

Orientation-invariant autoencoders learn robust representations for shape profiling of cells and organellesReviewer #1 (Remarks to the Author)

See attached.

Reviewer #1 Attachment on the following page

The authors have addressed my, and in my opinion, the other concerns, and made the manuscript stronger. In my opinion, the paper should be accepted for publication, highlighting the importance of orientation invariance to learned shape representations.

I have one major issue that should be determined by the authors and editor. It is now emphasized, multiple times, in the rebuttal and in the manuscript, that the focus of this manuscript is shape. The non-shape analyses are not very impressive and further work is required to support this point. Thus, I highly recommend revising the manuscript title to “shape” instead of “morphologic”, and adjusting the rest of the manuscript accordingly, bringing texture as a preliminary result toward future studies.

Minor edits and suggestions:

- Page 2, line #84: The hiPSCs and HPA datasets are referenced, but are not described. Readers unfamiliar with these datasets will appreciate an introduction. Same goes to how error is estimated. Both are described later in the text, but in my opinion should appear upon first introduced.
- Figure 2d legend, the analysis is shown per group without error bars. Can they be performed at the single cell / well level to provide statistics? (probably I am missing here something).
- Page 5, line #140, the panel should be 4b and not 4d.
- Page 6, line #160-166, if the authors want to be more thorough I would recommend normalizing by the organelle area for organelle contact analysis. This is not necessary to support the main contribution of this manuscript.
- Page 13 lines #420-423, the explanations in the Methods for selecting “k”s for k-means clustering should be elaborated. It was not clear to me how this decision was made based on the current text.

Assaf Zaritsky,

Ben-Gurion University of the Negev, Israel

Reviewer #2 (Remarks to the Author)

The reviewer still thinks that testing and comparing the of the proposed method with more controllable ground truth (i.e. such as comparing the embedding errors of the same cell morphology different added orientations) is necessary to demonstrate the performance of the method in addition to the real data set that the authors used.

The second comment raised by the reviewer is also not addressed in the revised manuscript. The authors still referred the observed spike as filopodia without providing molecular /structural evidence. Filopodia like structure should be used instead. The filopodia like structure that author identified does not directly from the O2-VAE prototype representation but from the morphology of individual cell grouping within the subtypes. The prototype level representation is similar to the previous cited paper and hence its unclear if O2-VAE is needed to identify this filopodia structure at single cell level. Comparison the results from applying the VAE or PCA is needed to support the claim.

Reviewer #4 (Remarks to the Author)

The authors have adequately addressed the concerns raised by Reviewer 3.

We make point-by-point responses to reviewers 1 and 2, and we note that reviewer 4 (in place of reviewer 3) did not request further revisions. We repeat the author's comments in italics, and our responses are in regular font. The changes to the manuscript and supplementary have been highlighted in yellow.

Reviewer #1

The authors have addressed my, and in my opinion, the other concerns, and made the manuscript stronger. In my opinion, the paper should be accepted for publication, highlighting the importance of orientation invariance to learned shape representations

- We thank the reviewer for their continued enthusiasm for the work and for their constructive feedback, which have strengthened the paper.

I highly recommend revising the manuscript title to “shape” instead of “morphologic”, and adjusting the rest of the manuscript accordingly, bringing texture as a preliminary result toward future studies.

- We agree with this suggestion and have changed the manuscript title to “Orientation-invariant autoencoders learn robust representations for shape profiling of cells and organelles.”
- In our previous revision, we changed the Abstract, Introduction, and Results to better emphasise that our work focuses on shape, rather than texture. In this revision, we have made further changes to the abstract, introduction, and main text to reiterate the focus on shape (the changes are highlighted in yellow). The main text occasionally uses the term morphology to avoid repetitive sentences and because it is appropriate as a biology term that indicates the size, shape, or structure of organisms or organelles.
- We have revised the main text to describe texture results as “preliminary”: lines 52, 136, 153 (highlighted in yellow).

Minor edits and suggestions:

- Thank you for raising these issues. We have made updates and highlighted the relevant changes in yellow, in particular:
 - *Page 2, line #84: The hiPSCs and HPA datasets are referenced, but are not described. Readers unfamiliar with these datasets will appreciate an introduction. Same goes to how error is estimated. Both are described later in the text, but in my opinion should appear upon first introduced.*
 - We have added a brief explanation of the hiPSC and HPA datasets in line 94. We have expanded the explanations for embedding errors and cluster consistency errors in the Results, starting line 88.
 - *Page 13 lines #420-423, the explanations in the Methods for selecting “k”s for k-means clustering should be elaborated. It was not clear to me how this decision was made based on the current text.*
 - We have added more details on the process for choosing the number of clusters in line 441. The brief summary is that we recommend visualizing

many samples from each cluster and choose a `k` that has low intra-class shape variability. Then, order the groups so that similarly-shaped groups are visualized next to each other (as we do in Fig.4). This requires judgement from the experimenter.

- *Figure 2d legend, the analysis is shown per group without error bars. Can they be performed at the single cell / well level to provide statistics? (probably I am missing here something).*
 - Thank you for requesting clarification since we would like this to be clear to the readers. Fig. 2 shows proportion data representing the fraction of errors out of ~400-500 pairs of cells ($n \sim 400-500$ for each of hIPSC, A549, A431, etc.). Pairs were determined by cross-correlation of the cell mask and manually confirmed to be different cells with similar shapes. The pairs are either profiled by pre-align + VAE or O2-VAE and results were counted as either correct or error for each pair.
 - Update to Fig 2f: We use nonparametric bootstrapping to compute 95% confidence interval estimates for the error rates. We perform a Chi-square test to determine whether the error rates are significantly different (for embedding errors, p-value < 0.001 for both hIPSC and aggregate HPA; for clustering errors, p-value = 0.11 for hIPSC and $p < 0.01$ for aggregate HPA). These analyses are referenced in the results section in line 96 and line 101. We elaborate on these in the Methods, starting 413.
 - We feel the analysis of orientation sensitivity on synthetic cell data requested by reviewer 2 was important to include in Fig. 2. Due to space constraints, we have reorganized the presentation as described below.
 - Fig 2d and 2f have been reorganized into a single figure that compares embedding errors and clustering errors for O2VAE vs. pre-align VAE on two real datasets, hIPSC ($n=533$) vs. HPA ($n=1718$).
 - In order to compare hIPSC vs. HPA, we aggregate the data for the 4 cells from the HPA (A549, A431, SK-MEL-30, and SiHa) to get the error rate for the HPA dataset. This is justified since they are from the same experimental dataset, and the goal is to show that VAE errors persist in real datasets even after pre-alignment.
 - The original Fig 2d and 2f with individual HPA cell lines are moved to the supplementary 2i for the interested reader. These figures now include 95% confidence interval estimates for the error rates.
 - This restructuring allowed for a new panel Fig 2b to demonstrate the orientation sensitivity of VAEs on synthetic cell data.

Reviewer #2

The reviewer still thinks that testing and comparing the of the proposed method with more controllable ground truth (i.e. such as comparing the embedding errors of the same cell morphology different added orientations) is necessary to demonstrate the performance of the method in addition to the real data set that the authors used.

- Thank you; this is an excellent suggestion. We have used our newly developed synthetic dataset Profiling Cell Shape and Texture to rigorously show that conventional VAEs suffer from orientation sensitivity and that our method O2VAE significantly reduces orientation sensitivity (Fig 2b, Supp Figure 2K).
- In the original Fig 2a, we did show the VAE embedding of ellipses and circles at different orientations and how this leads to poor clustering performance. We chose this example because it communicates the idea in a simple way. However, we acknowledge this is a toy dataset, and using synthetic cells would improve the argument.
- The new analysis is discussed in the Results from lines 77 to 83. We add a new chart in Fig.2b and explain the test in more detail in Supplementary 2H.
 - We take the synthetic dataset we generated for Figure 3, the “Profiling Cell Shape and Texture (PCST)” dataset. We make a randomly rotated copy for each image in the dataset and test whether they are clustered together when doing Kmeans clustering with k=10. A clustering error is defined as when orientation causes the same synthetic cell to be assigned to a different cluster.
 - Results:
 - Conventional VAEs show an overall clustering error rate of 58% compared to the O2VAE error rate of 7.8%. The O2-VAE error is not exactly 0 because there are discretization errors during the rotation transform as well as discretization in the encoder. See supplementary 2H for details.
 - The synthetic dataset allows probing of how orientation, eccentricity, and contour irregularity influence clustering errors. As expected, errors are significantly worse for rotation angles far from 0 or 180. We show this result in Fig.2b. There are also more errors with increasing eccentricity, which was also noted in the toy circle vs. ellipse dataset Fig.2a. Finally, conventional VAEs show increasing errors with increasing contour randomness. (Supplementary Fig.2k). These results provide evidence that clustering failures are due to orientation sensitivity and that O2VAE mitigates these errors.
- Fig. 2B (bars represent bootstrapped 95% confidence interval)

- Supp Fig. 2k (bars represent bootstrapped 95% confidence interval)

The authors still referred the observed spike as filopodia without providing molecular /structural evidence. Filopodia like structure should be used instead.

- The segmentations were based on Phalloidin-labeled immunofluorescence images of cells. Thus, F-actin is present in the long, thin, protruding structures. The public data

were collected elsewhere, and the LMNA +/- MEFs and fibronectin micropattern substrates are not readily available for further experiments.

- Importantly, the molecular identity of the thin protrusions has no bearing on the methods developed or the main conclusions. One of the main conclusions is that orientation invariance is important for learned shape representations, especially in autoencoder-based models, and this conclusion is unaffected by the change in wording.
- We have updated the manuscript to use “filopodia-like structure” instead of filopodia. The instances where we have changed filopodia to filopodia-like or filopodia-like structure are highlighted in yellow.

The filopodia like structure that author identified does not directly from the O2-VAE prototype representation but from the morphology of individual cell grouping within the subtypes. The prototype level representation is similar to the previous cited paper and hence its unclear if O2-VAE is needed to identify this filopodia structure at single cell level. Comparison of the results from applying the VAE or PCA is needed to support the claim.

- Thank you for raising this issue. The prior method, VAMPIRE, does not identify or report the finding of the filopodia-like structures in their results, prototype representations, or their discussion. However, we agree an empirical comparison is warranted and we have performed additional analysis.
- In Supplementary 3d, we use the original VAMPIRE algorithm to perform shape profiling and clustering. We randomly sampled from each VAMPIRE cluster to view examples of each class and after manual review, we see no enrichment of filopodia-like structures in any class. In contrast, random sampling of the O2VAE prototype shape classes followed by manual shows they are enriched for cells with thin filopodia-like structures (Fig.4a, clusters 2 and 7). We have also text to discuss the differences between the VAMPIRE and O2VAE prototype representations. In the main text in Line 150, we refer readers to the Supplementary 3d for additional discussion.

Reviewer #1 (Remarks to the Author):

The reviewers have addressed my concerns, I have no objection for publication.

Reviewer #2 (Remarks to the Author):

The reviewer thanks the authors for taking the time to address the reviewer's comments. I have no further question.